# Understanding the complexity of disease-climate interactions for rice bacterial panicle blight under tropical conditions

Johanna Echeverri-Rico[1], Eliel Petro[2], Paola A. Fory[2], Gloria M. Mosquera[2]*, Jillian M. Lang[3], Jan E. Leach[3], Juan D. Lobaton[2], Gabriel Garcés[1], Ricardo Perafán[1], Nelson Amezquita[1], Shirley Toro[1], Brayan Mora[2], Juan B. Cuasquer[2], Julián Ramirez-Villegas[2,4], Maria Camila Rebolledo[2,5], Edgar A. Torres[2¤]

1 Fedearroz–F.N.A. Bogotá, Colombia, 2 International Center for Tropical Agriculture (CIAT), Agrobiodiversity Research Area, Cali, Colombia, 3 Department of Agricultural Biology, Colorado State University, Fort Collins, CO, United States of America, 4 CGIAR Research Program on Climate Change, Agriculture and Food Security (CCAFS), c/o CIAT, Cali, Colombia, 5 CIRAD, UMR AGAP, F-34398 Montpellier, France. AGAP, Univ. Montpellier, CIRAD, INRA, Montpellier SupAgro, Montpellier, France

¤ Current address: RiceTec LTDA, Brazil
* g.m.mosquera@cgiar.org

**Data Availability Statement:** All relevant data are within the manuscript and its Supporting Information files.

## Abstract

Bacterial panicle blight (BPB) caused by *Burkholderia glumae* is one of the main concerns for rice production in the Americas since bacterial infection can interfere with the grain-filling process and under severe conditions can result in high sterility. *B. glumae* has been detected in several rice-growing areas of Colombia and other countries of Central and Andean regions in Latin America, although evidence of its involvement in decreasing yield under these conditions is lacking. Analysis of different parameters in trials established in three rice-growing areas showed that, despite BPB presence, severity did not explain the sterility observed in fields. PCR tests for *B. glumae* confirmed low infection in all sites and genotypes, only 21.4% of the analyzed samples were positive for *B. glumae*. Climate parameters showed that Montería and Saldaña registered maximum temperature above 34°C, minimum temperature above 23°C, and Relative Humidity above 80%, conditions that favor the invasion model described for this pathogen in Asia. Our study found that in Colombia, minimum temperature above 23°C during 10 days after flowering is the condition that correlates with disease incidence. Therefore, this correlation, and the fact that Montería and Saldaña had a higher level of infected samples according to PCR tests, high minimum temperature, but not maximum temperature, seems to be determinant for *B. glumae* colonization under studied field conditions. This knowledge is a solid base line to design strategies for disease control, and is also a key element for breeders to develop strategies aimed to decrease the effect of *B. glumae* and high night-temperature on rice yield under tropical conditions.

**Funding:** This research was supported by the Colombian Ministry of Science (formerly COLCIENCIAS), contract 556-2013, and the Global Rice Science Partnership (GRiSP). JR-V is supported by the Climate Services for Resilient Development (CSRD)-United States Agency for International Development (USAID) Award# AID-BFS-G-11-00002-10 toward the CGIAR Fund (MTO 069018), and the Climate Change, Agriculture and Food Security (CCAFS) project Agroclimas (http://bit.ly/2i3V0Nh). CCAFS is carried out with support from CGIAR Fund donors and through bilateral funding agreements (https://ccafs.cgiar.org/donors). For details, please visit https://ccafs.cgiar.org/donors.

**Competing interests:** The authors have no competing interests.

## Introduction

Global rice production is affected by abiotic and biotic factors and in some regions by a combination of closely related climate-host-pathogen interactions. The wide availability of historical data on yield and climate variables has allowed extensive analysis aimed at quantifying the role of climate in rice yield fluctuations, with temperature reportedly being one of the most important factors [1–4]. These results are due to the well-known detrimental effect of high temperature on rice during the reproductive stage, in which spikelet sterility is associated with increased temperatures above 35°C during flowering time [5, 6]. In addition, the effect of high temperature depends on its interaction with relative humidity [7], solar radiation [4], genotype, and management conditions (nitrogen), which explains why yield reduction due to climate change varies among regions experiencing similar temperatures.

High-temperature and -humidity conditions are also associated with yield reductions due to *Burkholderia glumae* infection, with reported yield reductions of up to 75% under high disease pressure [8]. *B. glumae* was first reported in Asia in 1956 and 1976 in association with grain and seedling rot, respectively [9, 10], and the corresponding disease cycle associated with the disease has been fully described. *B. glumae* disseminates through infected seeds and remains associated with below-ground plant tissue until the booting stage. At heading, the bacteria move to the aerial tissues and infect the panicles after emergence [11], producing rotting of grains of infected panicles. This infection process is favored by high relative humidity and rainfall during flowering stage [11, 12]. Different from Asia, *B. glumae* infection in the Americas is associated with spikelet sterility under high disease severity conditions, and in the Americas, the disease is known as bacterial panicle blight (BPB).

The optimal temperature for *B. glumae* multiplication is 30° to 35°C [13], a range that could produce grain sterility by itself on temperature-sensitive genotypes if the stress occurs during reproductive stage. However, artificial infections of rice with *B. glumae* under controlled conditions have shown that both grain rot and BPB can fully develop at 20° to 32°C [11, 14]. These previous reports also indicated a requirement for relative humidity above 95%, especially at the initial time of infection. Despite this previous research, uncertainty still exists about the ranges of temperature and humidity that are required for BPB development in field conditions under tropical environments, as well as whether *B. glumae* infection described in Asia for grain rot follows the same infection pattern to produce BPB observed in tropical regions of the Americas.

In 2007, BPB was observed in fields in Colombia, more specifically near the Caribbean coast in Montería and La Doctrina districts [15]. This was the first reported case in which *B. glumae* was associated with yield reduction since its first report in Colombia in 1989 [16]. During 2011, rice yield in Colombia decreased considerably and BPB was observed during the subsequent years. In fact, *B. glumae* was detected by several methods on BPB samples from fields in different rice production areas in Colombia [14], confirming the wide pathogen distribution in the country. However, estimation of disease incidence and severity, and its correlation with climatic parameters in each region, had not been assessed to date.

The capacity of tropical *B. glumae* strains to inhibit the grain-filling process was proven under controlled conditions [14], but field studies where disease and climate parameters were monitored to understand BPB disease under natural conditions had not been performed. For this purpose, trials were established in different rice-producing eco-zones and production systems in Colombia and a systematic sampling strategy was designed to study the relationships among disease parameters and plant responses in different environmental conditions. The results presented here will contribute to the design of different strategies aimed at decreasing the impact of BPB in rice production in tropical environments.

## Materials and methods

Plant tissue used in this study was collected at Fedearroz experimental stations by their own personnel. No collection permit was required.

### Molecular detection of *B. glumae*

*Burkholderia glumae*-specific primers `F-CGAAGGGTGTGGTTTGAACT` and `R-AACCTGCCA ACCTGTAATGC` were designed based on comparisons of genomic sequences from pathogenic strains from Colombia, Panama, Venezuela, and Costa Rica using the same strategy described for other bacterial pathogens [17, 18]. Primer specificity was tested using 10 ng of total bacterial DNA from strains of *Pseudomonas fuscovaginae* and *Acidovorax avenae*, 23 different *Burkholderia* species, *Xanthomonas oryzae* pv. *oryzae*, *Xanthomonas oryzae* pv. *oryzicola*, and *Pantoea agglomerans* (S1 Table).

Once the specificity of the set of primers was assessed using purified bacterial DNA, the same PCR conditions were used to evaluate performance on 15 rice panicles collected at different locations in Colombia that exhibited different degrees of disease, according to a 0 to 9 scale where 0 = no symptoms; 1 = 0.1–10.0% of the panicle affected; 3 = 11–20% of the panicle affected; 5 = 21–30% of the panicle affected; 7 = 31–60% of the panicle affected; 9 = >61% of the panicle affected. Each panicle was shattered and the seeds mixed to assure homogeneity. Two sets of 0.1 g of seeds were processed from each sample, one for DNA extraction and the other for to determine bacterial numbers on King's B medium as described by Fory et al. 2014 [14]. The 0.1 g sample for DNA extraction was dried for 24 h (EC Modulyo, EC Apparatus Inc., NY). Two stainless-steel ball bearings (2 mm) were added to the dried sample and plant tissue was disrupted on a TissueLyser (Qiagen, Hilden, Germany) at 30 Hz for 3 min. Total DNA was extracted from each powdered sample using Wizard® Genomic DNA Purification Kit Plus (Promega, Madison, WI), according to the manufacturer's protocol for plant genomic DNA. The PCR contained 10 ng of bacterial DNA or 100 ng of spikelet DNA, 7 μl of TAQ MIX (Promega, Madison, WI), and 2 μM of each forward and reverse primer in 15 μl final reaction volume. DNA amplification was performed in a thermal cycler (Mastercycler Nexus Gradient, Eppendorf, MA, USA) as follows: one cycle at 94˚C for 2 min; 29 cycles at 94˚C for 1 min, 62˚C for 1 min, and 72˚C for 1 min; and a final extension for 5 min at 72˚C. After amplification, 2.5 μl of the PCR product was resolved on 1.2% TBE agarose gel with 1.25 μl SYBR Safe (Invitrogen Co., Carlsbad, CA) by electrophoresis at 80 v for 90 min, and visualized by Gel Doc XR + SYSTEM (BioRad, Hercules, CA).

### Field study

**Locations, planting dates, and genotypes.** Three locations were selected to represent the main rice production areas in Colombia. A total of four genotypes and a local check were evaluated for disease-related parameters at each location. A local check represented a commercial variety adapted to that particular environment and currently being planted by farmers. Three different planting dates were scheduled in each location to capture as many climate-genotype-pathogen pressure combinations as possible. Details of all trials set for this study are shown in Table 1.

**Experimental design.** Trials were established during 2014 and 2015 to measure the effect of three factors on *B. glumae* infection: genotype (i = 5), location (j = 3), and planting date in each location (k/j = 3) and their interactions. Planting was performed under direct seeding with 60 kg of seeds/ha under a complete randomized block design with four replications and five genotypes at each site. Each experimental unit had 15 m$^2$ (3 m x 5 m), with 15 rows per genotype using 0.20 m space between rows. An integrated agronomic management was

**Table 1. Locations, planting dates, and genotypes used in this study.**

| Location | Planting Dates | Genotypes | Coordinates and Altitude |
|---|---|---|---|
| Montería, Córdoba | S1 (7/26/2014)<br>S2 (2/16/2015)<br>S3 (5/3/2015) | Fedearroz 2000<br>CT21375-F4-43-1<br>Fedearroz 50<br>IR64<br>Fedearroz 473 (local check) | 8,88666667 N<br>75,7911111 W 12 masl |
| Saldaña, Tolima | S1 (7/26/2014)<br>S2 (10/16/2014)<br>S3 (3/3/2015) | Fedearroz 2000<br>CT21375-F4-43-1<br>Fedearroz 50<br>IR64<br>Fedearroz 733 (local check) | 3,92916667 N<br>75,0155556 W 305 masl |
| Santa Rosa, Meta | S1 (7/23/2014)<br>S2 (5/8/2015)<br>S3 (7/9/2015) | Fedearroz 2000<br>CT21375-F4-43-1<br>Fedearroz 50<br>IR64<br>Fedearroz 174 (local check) | 4,1425 N<br>73,6294 W<br>467 masl |

S1, sowing date 1.

S2, Sowing date 2.

S3, sowing date 3.

applied in each location to have optimal plant nutrition and pest and disease control. Twenty experimental units (i.e., 5 genotypes times 4 replications) for each planting date (3) were analyzed per location (3), yielding in total 180 plots, for which four panicle developmental stages were used for sampling and analysis. The experimental design and sampling strategy are shown in Fig 1.

**Disease assessment on field study samples.** Sampling for measuring different disease-related parameters was performed as indicated in Fig 1. Two areas (0.20 m x 0.25 m) were marked in each plot and 9 panicles were randomly collected within each frame to obtain a composite sample formed by 18 panicles at booting, flowering, milky, and dough stage. Sampling time was adjusted for each genotype according to plant development. The booting stage was visually defined when 50% of the plants evidenced advanced panicle development inside the leaf sheaths. Flowering stage was registered when panicles of 50% of the plants had completed anthesis. Milky stage was calculated as 5 to 10 days after anthesis. Dough stage corresponded to the point when the grain was completely filled but without reaching physiological maturity. All four stage samples were used for bacterial detection by colony isolation and PCR, and incidence and severity were measured on milky-stage samples (Fig 1), following Eqs 1 and 2, respectively.

$$Incidence : \frac{\#panicles\ with\ symptoms}{Total\#of\ panicles\ evaluated\ (18)}\ x\ 100 \qquad [Eq\ 1]$$

After the evaluation of incidence, severity was registered using the same 0 to 9 scale described above in the molecular detection method. This information was used to calculate the severity index according to Eq 2:

$$Severity\ index = \frac{n(0) + n(1) + n(3) + n(5) + n(7) + n(9)}{Total\ of\ panicles\ evaluated} \qquad [Eq\ 2]$$

where $n$ indicates the number of panicles with each degree of damage (0 to 9) specified in parentheses.

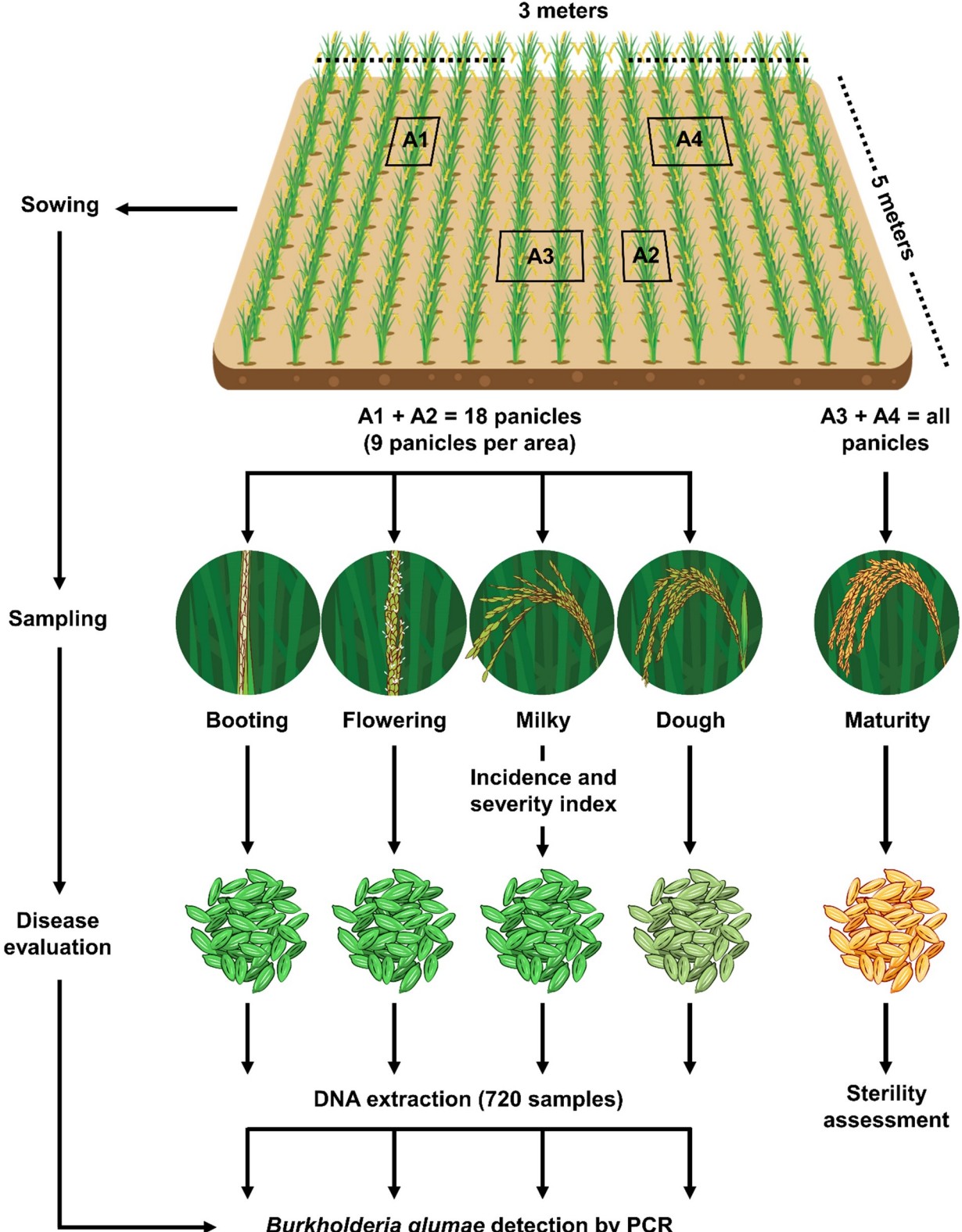

**Fig 1. Experimental design used for field trials.** Plot size and sampling strategy to measure disease parameters (A1 and A2, 0.05 m$^2$) and sterility (A3 and A4, 0.2 m$^2$).

In addition, another set of panicles was collected in two areas (0.40 m x 0.50 m) at physiological maturity for sterility assessment (Fig 1). For this evaluation, all spikelets recovered from matured panicles were manually sorted into two groups: unfilled that corresponded to empty spikelets and poorly filled grains and the other group fully filled grains. These values were used to calculate sterility by using Eq 3:

$$\%Sterility = \frac{\#of\ unfilled\ spikelets}{\#of\ total\ spikelets} x\ 100 \qquad [Eq\ 3]$$

Samples from the field study were processed as in the sensitivity test once incidence and severity were measured as indicated in Fig 1.

**Plant and disease data analysis.** Disease incidence, severity index, and sterility data were analyzed separately for each site to determine genotype responses in different seasons in the same location. Then, adjusted means were obtained from a combined analysis of variance that was performed to test the effect of genotype, season, and site with their interactions using the PROC GLIMMIX tool in SAS Studio v9.4 software. In the combined analyses of variance, blocks were considered as a random effect while genotype, season, and site were considered as fixed effects.

**Climate conditions and analysis.** Weather data were recorded in each trial using DAVIS Vantage Pro Plus weather stations (Davis Instruments, Hayward, CA). Recorded variables were precipitation, relative humidity, and maximum and minimum temperature, measured hourly during the entire plant growing cycle. Data were downloaded from the weather stations using WeatherLink software (Davis Instruments, Hayward, CA). Hourly data were aggregated to daily and from these five climate indices were computed over the 21-day period around flowering (i.e., starting 10 days before and ending 10 days after flowering). The calculated indices were number of days with maximum temperature above 34˚C (TMAX), number of days with relative humidity above 80% (HR), number of days with minimum temperature above 23˚C (TMIN), and number of days with precipitation above 30 mm (RAIN) during that particular 21-day period for each trial. Thresholds of temperature, precipitation, and relative humidity were chosen to reflect disease dynamics as well as rice heat stress [1]. Scatterplots helped explore some of the variability in the dataset to further disentangle climate-crop-disease relationships. Second, a correlation (Pearson) matrix was calculated between all of these parameters and incidence (INC) and sterility (STE) to detect any relationships between the disease and the weather conditions observed during 10 days before and 10 days after flowering. Incidence and sterility values used in the correlation analysis were averaged across all genotypes to produce a unique value per season.

## Results

### Molecular detection of *B. glumae*

**Detection of *B. glumae* using bacterial DNA.** New *B. glumae* primers (P11), were designed based on genomic comparisons of sequenced Latin American strains, and tested for specific amplification of an 834 bp fragment from only *B. glumae*, and not from other *Burkholderia* species (S1 Table. Fig 2). Similarly, no amplification was observed from other bacterial rice pathogens: *P. fuscovaginae*, *A. avenae* subsp. *avenae*, *P. agglomerans*, *X. oryzae* pv. *oryzae*, and *X. oryzae* pv. *oryzicola* (Fig 2),

**Sensitivity of *B. glumae*-specific primers using rice-infected seeds.** Collected rice seed samples used for primer sensitivity showed different disease severity levels, ranging from 0 to 9 (Table 2). PCR using 100 ng of total DNA extracted from 0.1 g of seed consistently detected bacteria in 10 samples with disease score ranging from 3 to 9 (Fig 3). Samples with disease scores 0 to 1 failed to amplify. *B. glumae* numbers were quantified from the same seed samples.

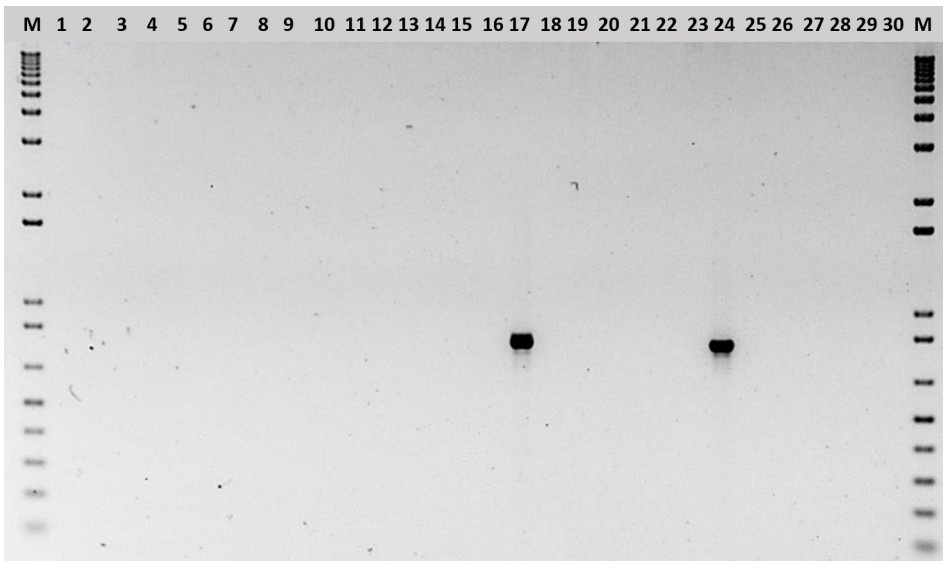

**Fig 2. Specificity test for P11 primers.** PCR Amplification from Different *Burkholderia* Species and Other Rice-Associated Bacteria. M, molecular marker 1 kb plus; 1, *B. phymatum*; 2, *B. cepacea*; 3, *B. tuberum*; 4, *B. graminis*; 5, *B. caledonica*; 6, *B. glatthei*; 7, *B. terricola*; 8, *B. xenovorans*; 9, *B. mimosarum*; 10, *B. sabiae*; 11, *B. diazotrophica*; 12, *B. symbiotica*; 13, *B. fungorum*; 14, *B. phenazinium*; 15, *B. phytofirmans*; 16, *B. vietnamiensis*; 17, *B. glumae*; 18, *B. gladioli*; 19, *B. kururiensis*; 20, *B. sacchari*; 21, *B. unamae*; 22, *B. tropica*; 23, *B. plantarii*; 24, *B. glumae 3252–8*; 25, *A. avenae* subsp. *avenae 4008–2*; *26, P. fuscovaginae 4500–2*; *27, Pantoea agglomerans TG7; 28, X. oryzae* pv. *oryzae PX0116; 29, X. oryzae* pv. *oryzicola BLS256*; 30, blank.

At least $10^5$ CFU/ml of *B. glumae* was detected in PCR-positive samples (Table 2), corresponding to samples with severity score higher than 3. Identical results were obtained in two independent experiments for bacterial quantification and DNA extraction.

**Table 2. Seed samples used to test diagnostic primer specificity and sensitivity.**

| Serial No. | Collection Site | [a]Disease Severity | PCR | [b]Bacterial Concentration (CFU/ml) |
|---|---|---|---|---|
| 1 | Ibagué, Tolima | 9 | + | $3.6\times10^5$ |
| 2 | Santa Rosa, Meta | 9 | + | $1.0\times10^8$ |
| 3 | Taluma, Meta | 1 | – | 0 |
| 4 | Libertad, Meta | 1 | – | 0 |
| 5 | Libertad, Meta | 1 | – | 0 |
| 6 | Libertad, Meta | 1 | – | 0 |
| 7 | Prado, Tolima | 9 | + | $9.0\times10^7$ |
| 8 | Santa Rosa, Meta | 9 | + | $2.0\times10^7$ |
| 9 | Purificación, Tolima | 9 | + | $1.2\times10^8$ |
| 10 | Purificación, Tolima | 9 | + | $1.2\times10^6$ |
| 11 | Purificación, Tolima | 3 | + | $1.0\times10^5$ |
| 12 | Purificación, Tolima | 3 | + | $2.0\times10^5$ |
| 13 | Palmira, Valle | 1 | – | 0 |
| 14 | Palmira, Valle | 0 | – | 0 |
| 15 | Palmira, Valle | 0 | – | 0 |

[a]Disease severity according to a 0 to 9 scale.

[b]Concentration of bacteria isolated from 0.1 g of infected seeds.

+Positive test.

-Negative test.

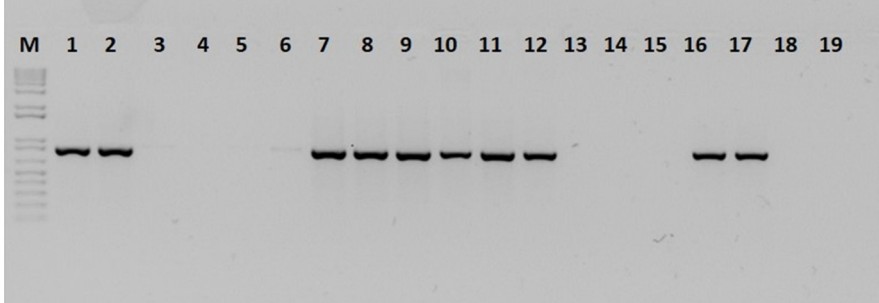

**Fig 3. PCR amplification with P11 primers using DNA from rice seeds with different levels of disease severity.** M, molecular marker 1 kb plus; 1 to 15, naturally infected rice seeds; 16, DNA *B. glumae* 3252–8; 17, DNA *B. glumae* 3252–8; 18, negative control (PCR reaction mix with water); 19, negative control (PCR reaction mix only).

## Disease evaluation in fields based on visual symptoms

Evaluation of panicles collected at the milky stage showed *B. glumae* symptoms in all three locations. In Montería, mean BPB incidence was similar during season 1 and season 3. Conversely, lower incidence was observed during season 2 (Table 3). At the genotype level, during season 1, disease incidence was very high in CT21375-F4-43-1, IR64, and the local check Fedearroz 473, but very low in Fedearroz 2000 and absent in Fedearroz 50 (Table 3). During season 3, again CT21375-F4-43-1, IR64, and Fedearroz 473 were the most affected genotypes and Fedearroz 2000 had less disease incidence, as was observed in season 1. Season 1 was the only one in which Fedearroz 50 did not show infection. Differences in incidence among genotypes and seasons were statistically significant, as was the interaction between genotype and season (Table 3). Disease severity index in Montería showed the same pattern as incidence, being higher during season 1 and season 3, which registered values similar to each other, and lower during season 2. These differences were statistically meaningful. Severity index was below 1.00 for all planting dates and genotypes, except for CT21375-F4-43-1, for which the severity index was 1.47 for the first planting date. Despite most values of severity index ranging from 0.00 to 1.00, the differences observed among genotypes and seasons were highly significant (Table 3). In Montería, although a high number of panicles showed disease symptoms (incidence), the proportion of affected grains was very low (severity index).

In Saldaña, BPB incidence was similar in seasons 1 and 3. In contrast BPB incidence was very low during season 2, which was planted in October 2015. In this location, most of the genotypes showed high incidence during season 1, with Fedearroz 2000 being less affected than the others. During season 3, incidence values were not significantly different among genotypes. Incidence for planting date 2 dropped in all genotypes (Table 3) and differences among seasons were highly significant, but not those among genotypes. Severity index at Saldaña showed CT21375-F4-43-1, IR64, and the local check Fedearroz 733, and, to a lesser extent Fedearroz 2000 and Fedearroz 50, as the most affected genotypes during season 1. During season 2, all genotypes showed similar severity index without significant differences. During season 3, however, the genotypes showed significant differences. CT21375-F4-43-1, Fedearroz 2000, and local check Fedearroz 733 were the most affected, while IR64 and Fedearroz 50 were significantly less affected by the disease (Table 3).

Santa Rosa had less disease incidence than the other two locations, and the degree of infection registered among genotypes was below 20%, except for Fedearroz 2000, which showed 70.37% during season 3, and Fedearroz 174 with 25% during season 2. For severity index, this site registered very low infection (Table 3).

**Table 3. Analysis of incidence, severity index, and sterility obtained during three different seasons and sites.**

| Site | Season | Genotype | Traits | | |
|---|---|---|---|---|---|
| | | | Incidence (%) | Severity index | Sterility (%) |
| Montería | S1 | CT21375 | 80.56 ± 49.27 a | 1.47 ± 0.47 a | 33.99 ± 2.84 ab |
| | | FED2000 | 3.70 ± 2.14 b | 0.04 ± 0.04 c | 25.11 ± 1.94 c |
| | | FED50 | 0.00 ± 0.00 b | 0.00 ± 0.00 c | 42.00 ± 4.30 a |
| | | IR64 | 81.94 ± 35.44 a | 0.82 ± 0.13 b | 26.32 ± 1.44 bc |
| | | FED473 | 61.11 ± 30.59 a | 0.46 ± 0.22 bc | 26.80 ± 0.95 bc |
| | | Mean | 29.68 ± 5.09 a | 0.56 ± 0.06 a | 30.52 ± 0.81 c |
| | S2 | CT21375 | 2.78 ± 1.45 a | 0.03 ± 0.03 a | 54.55 ± 0.54 b |
| | | FED2000 | 11.11 ± 5.09 a | 0.14 ± 0.11 a | 40.92 ± 2.29 c |
| | | FED50 | 4.17 ± 2.05 a | 0.04 ± 0.03 a | 78.25 ± 1.72 a |
| | | IR64 | 11.11 ± 5.04 a | 0.11 ± 0.04 a | 54.82 ± 3.85 b |
| | | FED473 | 1.39 ± 0.84 a | 0.01 ± 0.01 a | 53.36 ± 2.02 b |
| | | Mean | 7.46 ± 1.63 b | 0.07 ± 0.05 b | 57.07 ± 0.89 a |
| | S3 | CT21375 | 41.67 ± 18.14 ab | 0.50 ± 0.14 ab | 37.33 ± 1.17 a |
| | | FED2000 | 8.33 ± 3.85 c | 0.11 ± 0.04 c | 39.83 ± 0.97 a |
| | | FED50 | 25.00 ± 10.98 bc | 0.31 ± 0.08 bc | 36.28 ± 1.99 a |
| | | IR64 | 40.28 ± 17.55 ab | 0.54 ± 0.18 ab | 36.95 ± 1.65 a |
| | | FED473 | 54.17 ± 23.51 a | 0.79 ± 0.15 a | 37.09 ± 1.58 a |
| | | Mean | 32.37 ± 3.67 a | 0.45 ± 0.05 a | 37.52 ± 0.82 b |
| | Source of variation | G | *** | *** | *** |
| | | S | *** | *** | *** |
| | | G x S | *** | *** | *** |
| Saldaña | S1 | CT21375 | 79.17 ± 24.28 a | 0.79 ± 0.06 a | 14.05 ± 0.97 b |
| | | FED2000 | 23.61 ± 7.52 c | 0.24 ± 0.07 b | 8.13 ± 0.46 c |
| | | FED50 | 45.83 ± 14.23 bc | 0.54 ± 0.17 ab | 18.66 ± 1.98 a |
| | | IR64 | 64.81 ± 23.04 ab | 0.65 ± 0.08 a | 12.27 ± 3.09 b |
| | | FED733 | 63.89 ± 19.68 ab | 0.64 ± 0.11 a | 10.53 ± 1.64 bc |
| | | Mean | 56.73 ± 3.85 a | 0.57 ± 0.05 b | 12.30 ± 0.66 b |
| | S2 | CT21375 | 5.56 ± 2.05 a | 0.06 ± 0.03 a | 31.09 ± 2.61 ab |
| | | FED2000 | 9.72 ± 3.32 a | 0.13 ± 0.08 a | 29.10 ± 1.98 b |
| | | FED50 | 8.33 ± 2.90 a | 0.11 ± 0.09 a | 38.40 ± 3.72 a |
| | | IR64 | 8.33 ± 2.90 a | 0.08 ± 0.02 a | 31.67 ± 1.92 ab |
| | | FED733 | 9.72 ± 3.32 a | 0.10 ± 0.05 a | 29.99 ± 3.39 ab |
| | | Mean | 8.66 ± 1.72 b | 0.09 ± 0.04 c | 31.92 ± 0.92 a |
| | S3 | CT21375 | 83.33 ± 25.54 a | 1.08 ± 0.17 a | 35.52 ± 1.90 a |
| | | FED2000 | 59.72 ± 18.42 ab | 0.82 ± 0.10 ab | 31.61 ± 1.97 a |
| | | FED50 | 41.67 ± 18.35 b | 0.42 ± 0.03 bc | 35.33 ± 1.76 a |
| | | IR64 | 33.33 ± 12.08 b | 0.33 ± 0.17 c | 33.62 ± 1.31 a |
| | | FED733 | 83.33 ± 29.49 a | 0.91 ± 0.11 a | 33.63 ± 1.62 a |
| | | Mean | 62.87 ± 4.21 a | 0.71 ± 0.05 a | 33.99 ± 0.94 a |
| | Source of variation | G | ns | *** | *** |
| | | S | *** | *** | *** |
| | | G x S | ns | *** | ns |

(*Continued*)

**Table 3.** (Continued)

| Site | Season | Genotype | Traits | | |
|---|---|---|---|---|---|
| | | | Incidence (%) | Severity index | Sterility (%) |
| Santa Rosa | S1 | CT21375 | 6.94 ± 3.10 a | 0.07 ± 0.01 a | 25.69 ± 4.41 a |
| | | FED2000 | 12.50 ± 5.34 a | 0.13 ± 0.06 a | 28.67 ± 6.52 a |
| | | FED50 | 2.78 ± 1.40 a | 0.03 ± 0.02 a | 18.75 ± 5.97 a |
| | | IR64 | 3.70 ± 2.05 a | 0.04 ± 0.04 a | 22.28 ± 9.65 a |
| | | FED174 | 13.89 ± 5.90 a | 0.14 ± 0.05 a | 23.47 ± 6.66 a |
| | | Mean | 7.36 ± 1.38 b | 0.08 ± 0.02 c | 21.87 ± 1.64 b |
| | S2 | CT21375 | 20.83 ± 8.71 a | 0.35 ± 0.08 a | 38.64 ± 0.63 a |
| | | FED2000 | 2.78 ± 1.40 b | 0.03 ± 0.02 b | 34.97 ± 1.40 a |
| | | FED50 | 9.72 ± 4.22 b | 0.10 ± 0.03 b | 37.83 ± 1.26 a |
| | | IR64 | 5.56 ± 2.53 b | 0.06 ± 0.02 b | 38.80 ± 1.27 a |
| | | FED174 | 25.00 ± 10.39 a | 0.33 ± 0.04 a | 37.27 ± 2.48 a |
| | | Mean | 11.66 ± 1.80 a | 0.17 ± 0.02 b | 37.86 ± 1.93 a |
| | S3 | CT21375 | 1.39 ± 0.81 b | 0.01 ± 0.01 c | 33.09 ± 3.11 a |
| | | FED2000 | 70.37 ± 33.13 a | 1.00 ± 0.06 a | 34.16 ± 0.99 a |
| | | FED50 | 5.56 ± 2.92 b | 0.06 ± 0.03 bc | 35.44 ± 1.00 a |
| | | IR64 | 11.11 ± 4.78 b | 0.11 ± 0.02 b | 35.36 ± 2.41 a |
| | | FED174 | 8.33 ± 3.66 b | 0.08 ± 0.04 bc | 34.54 ± 2.59 a |
| | | Mean | 13.56 ± 2.24 a | 0.25 ± 0.02 a | 34.92 ± 1.90 a |
| | Source of variation | G | * | *** | ns |
| | | S | ns | *** | *** |
| | | G x S | *** | *** | ns |

G, genotype; S, season; G x S, genotype by season; CV, coefficient of variation. CT21375-F4-43-1: CT21375; Fedearroz 2000: FED2000; Fedearroz 50: FED50; Fedearroz 473: FED473. Means with the same letter are not significantly different from each other (P <0.05). Values represent means ± standard error.

Spikelet sterility at maturity varied by the geographic locations. In Montería, seasons 1 and 3 showed significantly lower sterility than season 2. Fedearroz 50 was the most affected during seasons 1 and 2, demonstrating its low adaptation to the conditions in this location. Meanwhile, Fedearroz 2000 showed the lowest sterility value for the same planting dates, suggesting tolerance of conditions found in Montería. Season 3 had similar sterility values for all genotypes and the differences were highly significant among genotypes, seasons, and their interaction (Table 3).

Saldaña conditions were less conducive to affecting the grain-filling process and lower levels of sterility were observed at this site than in Montería. Season 1 showed the lowest infection for the genotypes and only 12.30% was registered for the whole season. Sterility increased during seasons 2 and 3. At the genotype level, Fedearroz 2000 was the least affected during the three seasons, contrasting with the response of Fedearroz 50, which was the most affected. Differences in sterility among genotypes and seasons were significant.

In Santa Rosa, sterility values for all genotypes surpassed 20.0% during season 1, except for Fedearroz 50, which registered 18. 8%, but this value was not significantly different from those of other genotypes. An increase was observed during seasons 2 and 3, when sterility reached 37.9% and 34.9%, respectively, with all genotypes equally affected. Variation among seasons was highly meaningful, but variation among genotypes was not.

After the individual analysis of each parameter, a correlation analysis between incidence, severity index, and sterility showed no significant association. Thus, sterility was not associated with disease symptoms in any of the three locations analyzed (S1 Fig).

A combined analysis was performed with data derived from the four common genotypes in the three locations (eliminating the local checks). This analysis showed a significant effect (P <0.001) of genotype, season, and location and their respective interactions for the three characteristics evaluated: incidence, severity index, and sterility (S2 Table). A significant effect in the interaction of genotype, season, and location suggests that genotypes responded differently across season and location. Additionally, the combined analysis identified Saldaña as the location with the highest values for disease incidence and severity index, although sterility in this location was lower than in Montería and Santa Rosa. On the other hand, Montería was the location with the highest sterility, although lower values for incidence and severity index were observed. The same analysis showed that the pattern of the disease observed in each location was different according to incidence and severity index registered in all three seasons analyzed.

## Bacterial infection across locations, seasons

The sampling strategy used in the field study yielded 655 tissue samples from 720 expected because of limitations for tissue collection at some time points. For example, no samples were collected at flowering stage in Montería 2 and only one sample was collected in Santa Rosa 1 at booting stage. PCR and bacterial plating tests performed on the 655 samples showed that only 11 samples were positive for *B. glumae* using both methods, and 129 and 12 samples were positive using PCR or colony isolation, respectively (Table 4, S3 Table). A majority of the samples (503 out of 655) tested negative by both methods.

Although *B. glumae* was detected by PCR in a low number of samples, the results revealed the presence of the pathogen at all the sites and for the planting dates used in the study (S3 Table). Montería and Saldaña were the sites with the highest number of infected samples along the three different seasons. In total, 25, 17, and 25 samples tested positive by PCR for planting dates 1, 2, and 3 in Monteria, and 12, 19, and 23 in Saldaña (S3 Table). Conversely, Santa Rosa was the site with the least infection, with only 7, 9, and 3 positive samples for planting dates 1, 2, and 3 (S3 Table).

## Climate conditions during flowering time and their association with BPB

Disease-related variables and sterility (Table 3) were analyzed in relation to specific climate parameters. The analyses revealed that a wide variation occurred in weather conditions during the 21 days around flowering time across the trials carried out in this study (S4 Table). The number of days with high maximum temperature >34˚C varied from 0 to 16, and the high minimum temperature >23˚C varied from 1 to 20 days, indicating stressful conditions during the rice reproductive phase. Importantly, Montería and Saldaña sites experienced high minimum temperature conditions for more than 11 days for all planting dates, whereas Santa Rosa site had fewer days, from 1 to 3, with this condition. Thus, Montería and Saldaña were

**Table 4. Number of positive or negative samples for *B. glumae* using two detection methods.**

| Colony Isol. | PCR | # of Samples | % of Samples |
|:---:|:---:|:---:|:---:|
| P | N | 12 | 1.8 |
| P | P | 11 | 1.6 |
| N | P | 129 | 19.6 |
| N | N | 503 | 76.7 |

P, positive; N, negative

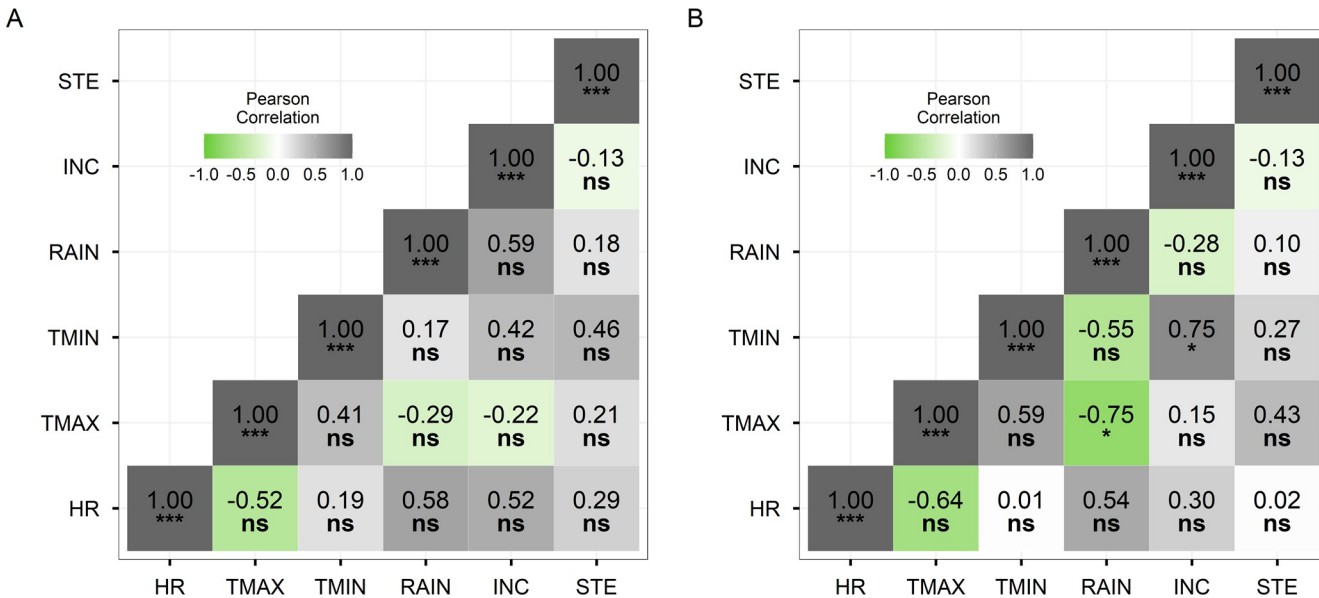

**Fig 4.** Pearson Correlation between Disease Variables and Climate Variables for 10 Days Before (A) and 10 Days After (B) Flowering Date. HR, relative humidity; TMAX, maximum temperature; TMIN, minimum temperature; RAIN, precipitation; INC, disease incidence; STE, spikelet sterility.

environments with higher temperatures than Santa Rosa. Relative humidity (RH) conditions varied substantially across both locations and planting dates. For instance, Santa Rosa registered 7, 15, and 21 days of RH above 80% during S3, S1, and S2, respectively. Montería showed 21 days with RH above 80% during S1 and S3, but only 13 during S2. In Saldaña, the number of days with RH above 80% were fewer than those registered in Montería and Santa Rosa, and S1 and S3 registered 11 and 12 days with this condition. S2 was the driest of all, registering only 8 days with maximum humidity above 80%. Conversely, the number of days with extreme precipitation (>30 mm) showed low variation, being consistently low across the trials, with a maximum of 4 days in S2 in Santa Rosa, and 2 or fewer days for the other locations and planting dates.

Despite variation in weather conditions during the 21 days around flowering, TMAX/RH, TMAX/RAIN, TMIN/RH, and TMIN/RAIN paired combinations were not significantly correlated with the disease parameters (S2 Fig). A slight clustering was observed between high incidence and more than 14 days with TMIN >23˚C, but not with TMAX 34˚C or RAIN >30 mm. The next step was to analyze the same climatic thresholds and disease parameters, but using two time windows, 10 days before (Fig 4A) and 10 days after flowering date (Fig 4B). Correlation analysis showed significant correlation between TMIN 23˚C (P <0.05) and incidence at 10 days after flowering. A negative correlation detected between TMAX and RAIN was expected since rain during daytime has a cooling effect on the environment. No correlation was found between climate and disease when analysis was performed on 10 days before flowering (data not shown).

## Discussion

High temperature from 32 to 36˚C during anthesis can induce spikelet sterility and limit grain yield in rice [19]. Similarly, the rice pathogen *B. glumae*, which optimally grows from 30 to 35˚C, can also induce spikelet sterility when infection occurs during anthesis[14]. Since 2009, decreases in rice grain yield have been reported in several regions of Colombia [20], and in

some cases BPB has been described as the main causal agent. To evaluate whether *B. glumae* disease is affecting yield in Colombia, experimental rice plots were established in three different rice-growing areas. Five genotypes were planted in each location at three different planting dates spanning two consecutive years, 2014 and 2015, and multiple variables were analyzed during the reproductive stage.

Visual symptoms were used to assess disease incidence and severity in each trial. Disease incidence has been widely used to design forecast models for BPB, develop automatic disease evaluation tools, identify disease resistance QTLs, and quantify the effect of the disease on yield [21–24]. The strategy to collect incidence data differs among studies, but this study opted for the milky stage since that is the time when BPB symptoms can be clearly differentiated from symptoms produced by other biotic or abiotic factors. Moreover, if symptom evaluation or pathogen detection is performed at the milky stage, the effect of disease infection on grain formation can be directly quantified and the relationship between disease and spikelet sterility can be established. Symptoms related to grain rotting or darkening were rarely observed in the analyzed samples and, when present, they were not taken into account for BPB rating since they could be due to multiple factors, including other bacterial and fungal infections.

Based on incidence data, BPB was found in all the sites and genotypes. Saldaña was the location with the highest incidence, followed by Montería. In contrast, conditions in Santa Rosa were less favorable for the disease. Besides incidence, evaluations were performed to quantify the degree of damage caused by the pathogen to each panicle, information processed as severity index. Grain weight losses associated with different severity grades could vary depending on the genotype according to previous studies conducted in Saldaña using a scale similar to the one reported here [25]. That study confirmed S1 season in Saldaña as favorable for high disease incidence, although the maximum severity obtained was 1.7. Under those conditions, grain weight loss in Fedearroz 2000 was about 10% and in Fedearroz 473 35% when BPB severity degree was 1.1 and 1.3, respectively. Although sterility was not measured, incidence and severity did not correlate with yield. However, the negative impact of disease severity on yield could depend on the genotype response. Taking the two studies together, it is clear that disease incidence is only an indicator of pathogen presence, and that yield is only impacted when panicles have damage above 20%, corresponding to severity index higher than 3. Despite these results, severity seems to be a more efficient measure of pathogen capacity to affect grain formation (sterility) or development (low grain weight), both related to yield loss.

Analysis of yield and weather datasets from two Colombian regions, including Saldaña, showed that rice yield is affected by unfavorable climate conditions such as high nighttime temperature in certain months of the year [26]. As expected, the degree of climate impact also depended on the genotype. The sterility we observed might have been caused by other factors. Therefore, relationships between disease severity values and yield under field conditions should be interpreted with caution, especially in cultivars susceptible to site-specific climate factors. Results from our study showed that disease severity was below 1 in all locations and seasons, except for CT21375 during S1 in Montería and S3 Saldaña. Conditions during the field trials were not favorable for higher damage; in fact, Saldaña, demonstrated higher disease incidence but the lowest level of sterility. According to our findings, disease incidence or sterility should not be used as the only parameters to evaluate pathogen impact on yield. Moreover, the use of incidence or sterility could overestimate disease pressure, thus compromising the accuracy of results, and potentially having negative implications for downstream applications in disease prediction models or germplasm selection for disease resistance. Nonetheless, disease assessment at the milky stage and the use of severity should be considered for future studies on BPB.

*B. glumae* has been detected on asymptomatic rice tissues [14, 27], which suggests that bacterial growth can take place without inducing symptoms on the rice seed coat. Whether this type of colonization could still induce spike sterility is not yet known and, if this is feasible, a low severity index does not necessarily imply low bacterial infection. For this reason, we designed a more specific method to confirm bacterial infection in the absence of visual symptoms. PCR results were consistent with observed low disease severity, suggesting no bacterial infection, since only 21.4% of the analyzed samples tested positive for *B. glumae*. Most of the positive samples were collected in Montería and Saldaña, with few of them in Santa Rosa. A study in the United States reported *B. glumae* on 25% of the panicles showing BPB symptoms [28], which suggested alternative causes for the evaluated symptoms or levels of infection below the detection limit for bacterial isolation on culture media. In our case, lower infection in the field yielded few positive samples by both PCR and isolation, although PCR showed a higher number of samples with *B. glumae*. Our PCR method detected bacterial infection above $10^5$ CFU per 0.1 g of plant tissue; the detection limit might improve by extracting DNA from more tissue. We used 0.1 g because that was the maximum amount that could be processed by an automatic tissue grinder, a method that facilitates processing a high number of samples in a short period of time. In addition, our method could be practical for labs processing a high number of samples since direct DNA extraction using an automatized system diminishes labor requirements. Despite the advantage of PCR over colony isolation, we did find 12 samples out of 655 tested that were positive by plating but where no PCR amplification was observed.

According to available information, bacterial colonization affecting emerged panicles is developed before panicle emergence [29]. Although a low number of infected samples was expected from severity index data, PCR might be used in future studies to investigate the dynamics of infection through different panicle developmental stages.

BPB has been associated with warm temperatures and frequent rainfall during flowering time, conditions that were used to develop and test a model to forecast BPB infection using the number of days with RH above 80% and the number of days with minimum temperature above 22˚C [21]. Shew et al. (2019) [30] used the same indicators to model *B. glumae* impact on rice grain production in the United States. Here, TMAX >34˚C, TMIN >23˚C, RH >80%, and RAIN >30 mm indicators during flowering time were not clearly associated with either incidence or severity index, suggesting that the climatic thresholds for high BPB in the tropical region may be different from those defined for temperate environments. However, infection was greater at irrigated sites (Saldaña and Montería) than at the rainfed site (Santa Rosa), suggesting that management may have played a role in obscuring the pathogen-crop-climate relationships. This is consistent with previous studies that indicate that frequent low-intensity precipitation events increase *B. glumae* infection and its impact on rice [11]. However, more studies are needed to confirm the extent to which BPB disease is indeed influenced by management and its interaction with local climate conditions and crop microclimate.

The results obtained in this study were not conclusive about a direct involvement of *B. glumae* as the causal agent of sterility. Besides climate the presence of "unknown" disease tolerance in the genotypes used in this study could be an additional cause of the low severity registered in all trials. In fact, Fedearroz 2000 has been identified as a cultivar susceptible to *B. glumae* under greenhouse conditions using different strains isolated from Montería and Saldaña, which produce damage on inoculated panicles from 23% to 100% [31]. Based on this information, we treated Fedearroz 2000 as a susceptible check in the field trials. Low severity observed under natural conditions, including for Fedearroz 2000, suggests a tight interaction between *B. glumae*-rice-environment. This complex interaction should be considered before dismissing the potential of this bacterium to decrease yield in more susceptible genotypes.

High night temperatures or high humidity are considered favorable conditions for BPB development [8, 32, 33], but this description is still quite imprecise. Our study provides detailed information about rainfall and minimum and maximum temperatures during 10 days prior to and after flowering time. Montería and Saldaña registered TMAX above 34˚C, TMIN above 23˚C, and RH above 80%, from which only TMIN correlated with disease incidence. Therefore, this correlation and the fact that Montería and Saldaña had a higher level of infected samples according to PCR tests, high minimum temperature but not maximum temperature, seems to be determinant for *B. glumae* colonization under field conditions.

The results presented here show that the infection pattern found in Colombia does not follow the invasion model described for this pathogen in Asia and the requirement for more drastic minimal temperatures above 23˚C [22] to have severe infections is knowledge that establishes a solid baseline for future research on this disease under the tropical conditions of America. However, future studies would be improved by the identification and use of a more susceptible cultivar, the inclusion of more seasons, and the performance of independent analysis for irrigated and rainfed environments.

For management strategies, prediction of the disease has been a recurrent theme among farmers and extension services and a forecast model for *B. glumae* disease has been described [21]. Similarly, an imaging-based technology to screen rice seeds infected by *B. glumae* has been published [23]. Both tools rely on the association of seed symptoms with bacterial infection, but the reliability of this association may depend on geographic region. Symptoms produced by *B. glumae* on rice grains have been widely described in Asian countries as grain discoloration and grain rot [34], and there is a possibility they could vary from those symptoms observed in tropical rice production areas such as those in Colombia. In Colombia, *B. glumae* disease in rice panicles is mainly recognized by straw-colored spikelets, sometimes accompanied by dark-brown patterns located at the base, middle, or upper tip of the seed [14], symptoms that most of the time are restricted to the seed coat without affecting the color of the grain. Results obtained by Mulaw et al. (2018) [28] suggest that disease evaluation based on visual symptoms is imprecise, consistent with our results. Together, these studies highlight the necessity for additional efforts to refine the current epidemiological model and the available disease evaluation methods for BPB. Moreover, the environmental profiles during flowering and early grain-filling processes reported here could contribute to the refinement of climate profiles conducive to high disease pressure under tropical conditions.

The effect of high temperature or *B. glumae* at flowering time could be complex if both stressors overlap in the same field. Molecular diagnostics would complement field assessment in fields affected by any abiotic limitation. In addition, management strategies should focus on developing varieties with both traits, *B. glumae* resistance and high night-temperature tolerance.

## Supporting information

**S1 Fig.** Correlation between Disease Incidence (A), Severity (B), and Spikelet Sterility in Three Locations.
(TIF)

**S2 Fig. Climate conditions registered during 21 days' time frame during flowering time at the three sites for all planting dates and their relationship with observed disease incidence and severity.** Numbers beside symbols indicate the planting date (1–3, as specified in Table 1). Panels show pairs of climate variables as follows: (A) TMAX vs precipitation; (B) TMAX vs HR; (C) TMIN vs precipitation; and (D) TMIN vs RH. All panels show severity (in colors) and

sterility (symbol size).
(TIF)

**S1 Table. Bacterial species used to test P11 primers specificity.**
(PDF)

**S2 Table. Combined analysis of variance for incidence, severity index, and sterility of four rice genotypes across nine environments.**
(PDF)

**S3 Table. *B. glumae* detection on rice samples collected from field study using colony isolation and PCR.**
(PDF)

**S4 Table. Climate conditions registered during 21 days' time frame during flowering time.**
(PDF)

## Acknowledgments

The authors acknowledge Alex Gozalez (Fundación DANAC, Venezuela) and Luis Vargas (INTA, Costa Rica) for providing *B. glumae* strains for sequencing analysis. We also knowledge Dr. Gilles Bena (IRD, France) for providing DNA from *Burkholderia* species, and Koss Kini for donating DNA from *P. agglomerans*. The views expressed in this article cannot be taken to reflect the official opinions of these organizations.

## Author Contributions

**Conceptualization:** Gloria M. Mosquera, Maria Camila Rebolledo, Edgar A. Torres.

**Data curation:** Johanna Echeverri-Rico, Eliel Petro, Maria Camila Rebolledo.

**Formal analysis:** Johanna Echeverri-Rico, Eliel Petro, Juan D. Lobaton, Brayan Mora, Juan B. Cuasquer, Julián Ramirez-Villegas, Maria Camila Rebolledo.

**Funding acquisition:** Gloria M. Mosquera, Maria Camila Rebolledo, Edgar A. Torres.

**Investigation:** Eliel Petro, Gloria M. Mosquera, Jan E. Leach, Julián Ramirez-Villegas, Maria Camila Rebolledo, Edgar A. Torres.

**Methodology:** Johanna Echeverri-Rico, Eliel Petro, Paola A. Fory, Gloria M. Mosquera, Jillian M. Lang, Gabriel Garcés, Ricardo Perafán, Nelson Amezquita, Shirley Toro, Julián Ramirez-Villegas.

**Project administration:** Gloria M. Mosquera.

**Resources:** Jan E. Leach.

**Supervision:** Gloria M. Mosquera.

**Visualization:** Eliel Petro.

**Writing – original draft:** Johanna Echeverri-Rico.

**Writing – review & editing:** Gloria M. Mosquera, Jillian M. Lang, Jan E. Leach, Juan D. Lobaton, Gabriel Garcés, Julián Ramirez-Villegas, Maria Camila Rebolledo, Edgar A. Torres.

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
