## [Decision Letter · Decision Letter 0]

12 Feb 2021

PONE-D-21-01039

Understanding the Complexity of Disease-Climate interactions for Rice Bacterial Panicle Blight under Tropical Conditions

PLOS ONE

Dear Dr. Mosquera,

Thank you for submitting your manuscript to PLOS ONE. After careful consideration, we feel that it has merit but does not fully meet PLOS ONE’s publication criteria as it currently stands. Therefore, we invite you to submit a revised version of the manuscript that addresses the points raised during the review process.

Although both reviewers found your work to be suitable for publication, they also raised the following concerns about the manuscript. The language in the manuscript needs improvement and authors are advised to avoid long sentences to improve clarity. Another concern of the reviewer is the quantification of pathogen based on PCR and alternatively isolation and quantification of bacteria was suggested. The interpretation should be revised

We look forward to receiving your revised manuscript.

Kind regards,

Prasanta K. Subudhi, Ph.D.

Academic Editor

PLOS ONE

Additional Editor Comments:

Major revision

Journal Requirements:

Reviewers' comments:

Reviewer's Responses to Questions

**Comments to the Author**

1. Is the manuscript technically sound, and do the data support the conclusions?

Reviewer #1: Partly

Reviewer #2: Yes

2. Has the statistical analysis been performed appropriately and rigorously? 

Reviewer #1: Yes

Reviewer #2: Yes

3. Have the authors made all data underlying the findings in their manuscript fully available?

Reviewer #1: Yes

Reviewer #2: Yes

4. Is the manuscript presented in an intelligible fashion and written in standard English?

Reviewer #1: Yes

Reviewer #2: Yes

5. Review Comments to the Author

Reviewer #1: The manuscript titled “Understanding the Complexity of Disease-Climate interactions for Rice Bacterial Panicle Blight under Tropical Conditions” by Echeverri-Rico et al, investigates Bacterial Panicle Blight under Tropical conditions, specifically in three geographic areas of Colombia. The manuscript presents data surveying BPB disease parameters in three geographic areas in Colombia, using 5 different rice genotypes in three seasons. The field experimental design and data analysis is flawless, and the amount of field data collected is significant. The study found that two of the regions as particularly conducive for disease development, unfortunately, it appears that against the expectations, the effect of climate on disease was not observed. Moreover, although BPB has been associated with sterility, this study contradicts previous observations. What the study did found, was the interaction between genotype and location. This finding is not particularly surprising as resistance to BPB is a quantitative trait, as previously reported. Overall, while the field data is valuable, it is merely descriptive.

My main concern with this study is related to the analysis of the presence of the pathogen using PCR. The study tries to provide explanation on disease parameters based on their ability to detect bacteria by PCR. PCR is not appropriate to do that due to technical limitations extracting DNA from particular tissues or genotypes, or the fact that PCR can generate a false positive, amplifying DNA from dead bacteria that are not actively contribute to infection. The best test to determine active infection is by isolation and quantification of bacteria. In addition, the text does not clearly explain (I apologize if I missed it), which gene was used to design PCR primers. Is it really B. glumae specific? There are many species of Burkholderia in natural environments.

In addition, the study makes grandiose interpretations when the percentage of positive results is very low, and the experiments lack appropriate controls to determine if the results are biologically meaningful, or associated with technical problems. For example, statements such as: “Disaggregated analysis by location evidenced Montería as the site with the highest number of infected samples and where the bacterial infection along three different seasons was quite stable” (Line361-363), begs the question, what is the criteria to consider that scenario “stable”, just the ability to detect bacteria? The levels of detection of bacteria are too low to support such a bold statement.

Line 371: States “PCR results were used to elucidate the dynamic of bacterial infection through panicle development”. That statement indicates a lot more than what the data actually shows. The data exclusively shows sampling at different developmental stages, does not necessarily provides mechanistic insight into the “dynamics of bacterial infection”. Dynamics of infection will mean having the ability to monitor disease progression by tracking bacteria throughout the tissues, which is not what they did.

Since the field data is solid and well analyzed and the PCR data does not add any particular insight, I would recommend remove all the PCR data.

Minor comments:

Line 32: Change to “under the conditions studied”. Is that statement referring to the three sites? It seems that the levels of inoculum are different among the sites.

Line 61: citations are incorrect for that fact.

Line 61: several years ago is very vague statement for a scientific publication.

Lines 61-63: not clear what do the authors mean to say.

Table 1. I recommend to add dates in format (month/day/year).

Line 167: is it showed differences?

Line 276: Please be consistent with the use of terms, “sowing” and “planting” is used interchangeable.

Line 509: Edit to “It is clear”

Line 641. Please change to Dr. Jan Leach.

Other comments:

Figure legends are somehow inserted in the text, interrupting the flow of the narrative.

Reviewer #2: This manuscript reports experimental data showing the effects of rice genotype, culture location and climate condition on the occurrence of bacterial panicle blight in Columbia. I think this study has merits to be published in this journal because of its comprehensive field data to understand the environmental factors influencing the disease. Nevertheless, this manuscript needs to be improved significantly for better deliverance of information to readers.

1. The Results part needs an elaborated editing process to make sentences shorter and clearer. Overall, it tends to be too lengthy describing too much details of data presented in Table 2. English writing itself also looks rough in the Results part compared with other parts including introduction, so additional round of internal review process is recommended.

2. Lines 421 – 444: Please cite the corresponding tables and figures after each sentences.

3. Fig. 4B: The significant parts, RAIN x TMAX (-0.75) and TMIN x INC, should be addressed in the result and discussion parts in more detail.

4. Other specific comments:

a) Line 61: 1956 is not several years ago

b) Line 167: English (please recheck grammar and correct errors)

c) Line 183: Should be ‘Equation 2’

d) Line 247 and other places: Please use ‘period’ for decimal number throughout the text.

e) Table 2: I suggest Table 2 in the landscape orientation for easier reading

f) Line 276: sowing 2 should be season 2, drop should be dropped

g) Line 337: English (‘and’ should be removed?)

h) Line 341: English (…interaction, genotype x …)

i) Line 386: remove ‘in’

j) Figure 4 legend: Please explain each label/code in the figure, e.g. STE, INC, HR

k) Line 494-496: not understood well. Please explain more detail..

l) Line 507-511: Not understood.. Please rewrite

m) Line 533: should be 24

n) Line 547: Highly infected

6. PLOS authors have the option to publish the peer review history of their article (what does this mean?). If published, this will include your full peer review and any attached files.

Reviewer #1: No

Reviewer #2: No

---

## [Author Response · Author response to Decision Letter 0]

30 Mar 2021

Dear reviewers. Thank you so much for your valuable comments that will help in improving the manuscript.

Every comment received in the decision email was address in the document and also is described in the document "Response to reviewers".

---

## [Decision Letter · Decision Letter 1]

20 Apr 2021

PONE-D-21-01039R1

Understanding the Complexity of Disease-Climate Interactions for Rice Bacterial Panicle Blight under Tropical Conditions

PLOS ONE

Dear Dr. Mosquera,

Thank you for submitting your manuscript to PLOS ONE. After careful consideration, we feel that it has merit but does not fully meet PLOS ONE’s publication criteria as it currently stands. Therefore, we invite you to submit a revised version of the manuscript that addresses the points raised during the review process.

Although the manuscript has been improved substantially, the reviewers have raised some concern regarding your conclusion of disease severity as a parameter for grain sterility or reduction in grain yield. The authors should provide some plausible explanation for this discrepancy.

We look forward to receiving your revised manuscript.

Kind regards,

Prasanta K. Subudhi, Ph.D.

Academic Editor

PLOS ONE

Journal Requirements:

Additional Editor Comments (if provided):

Minor revision

Reviewers' comments:

Reviewer's Responses to Questions

**Comments to the Author**

1. If the authors have adequately addressed your comments raised in a previous round of review and you feel that this manuscript is now acceptable for publication, you may indicate that here to bypass the “Comments to the Author” section, enter your conflict of interest statement in the “Confidential to Editor” section, and submit your "Accept" recommendation.

Reviewer #3: All comments have been addressed

Reviewer #4: (No Response)

2. Is the manuscript technically sound, and do the data support the conclusions?

Reviewer #3: Partly

Reviewer #4: Partly

3. Has the statistical analysis been performed appropriately and rigorously? 

Reviewer #3: Yes

Reviewer #4: Yes

4. Have the authors made all data underlying the findings in their manuscript fully available?

Reviewer #3: Yes

Reviewer #4: (No Response)

5. Is the manuscript presented in an intelligible fashion and written in standard English?

Reviewer #3: Yes

Reviewer #4: No

6. Review Comments to the Author

Reviewer #3: Echeverri-Rico et al. present the investigation of environmental and genotype effects on bacterial panicle blight incidence and severity, and possible connection to sterility. The results provide important information that will improve our understanding of factors that contribute to BPB development and will be useful for the formulation of improved management strategies. Overall, the experiments were well defined and executed. Although relatively minor, I do not agree with one conclusion stated in the discussion regarding the use of severity as a parameter to quantify impact on yield (see comment below). Additionally, I have outlined several minor suggestions and comments below.

L90-91: Should be “..where disease and climate parameters were monitored…”?

L138: change to “a local check”

Figure 1: change to “DNA extraction”

L252: change to “consistently”

L255: A little confusing as written. It may be more clear if written as “At least 10^5 CFU/mL…” or something similar.

Figure 2: The lane numbers are slightly misaligned in some cases.

L283-284: Based on Table 3, it appears that Fedearroz 50 had infection in Seasons 2 and 3, not Season 1?

L304-307: This is slightly misleading. Although CT21375-F4-4301, IR64, and Fedearroz 733 have the highest mean incidences, Fedearroz 50 is not significantly different than IR64 and Fedearroz 733. I suggest re-wording to clarify this.

L319: Change to “and the”

L371: Should this be 129 and 12 samples? The 11 samples that tested positive via both methods were referenced previously should not be added to the 129 samples that only tested positive with PCR.

L366-373: Although the overall number of positive samples was low, was there any indication that a particular sampling timepoint produced more positive samples than others? Perhaps pathogen presence was too low at the earlier sampling dates, but was sufficient at later dates/stages?

L462-464: Mean severity was also greater than 1 for CT21375 in S3 in Saldana.

L467-468: What evidence is provided to make this conclusion? Based on results presented in Table 3, it does not appear that severity is correlated with sterility. For example, in the mentioned example of CT21375 in Monteria S1, which had the highest severity, its sterility was not significantly different than FED50, which had no recorded infection. Other examples are shown where differences in severity were observed, but no differences in sterility.

Reviewer #4: The manuscript aimed to improve our understanding of the interaction between climate and Bacterial Panicle Blight disease infection in tropical environment. Although authors made some improvement in the revised manuscript, it still needs further revision due to the following:

There are some places in the discussion, the authors statements are contradictory. For example (lines 458-462), grain weight loss in Fedearroz 2000 is about 20% and is 50% in Fedearroz 473 when BPB severity degree is around 1 (Table 3). These lines showed very low sterility. Under this circumstance, author’s conclusion to use severity as a precise parameter for quantifying pathogen capacity to affect grain development or yield loss is nor based on the results obtained in this manuscript. The authors should take note of the fact that if the genotypes are resistant to infection, they should provide an explanation why there is increased sterility and reduced grain yield despite low infection and low severity.

In Line 517, authors admitted that ‘the results obtained in this study were not conclusive about a direct involvement of B. glumae as the causal agent of sterility’.

Some minor comments:

Line-51: Delete ‘affecting yield’

L-65: change ‘on’ to ‘of’

L-74: indicate to indicated

L-119 Fory et al. 2014. REFERENCE NUMBER???

L-138: alocal to a local

L-191 deficiently to poorly

L-251: 0 to 9 to score 0 to 9; Delete ‘according to the 0 to 9 scale’

L-252: consistently to consistently

L-256: Delete ‘on the 0 to 9 scale’

L-290: 1.47 was observed for the first planting date: change to ‘severity index was 1.47 for the first planting date’

Fig 2 need improvement with line alignment.

7. PLOS authors have the option to publish the peer review history of their article (what does this mean?). If published, this will include your full peer review and any attached files.

Reviewer #3: No

Reviewer #4: No

---

## [Author Response · Author response to Decision Letter 1]

6 May 2021

Reviewer #3 specific comment. L366-373: Although the overall number of positive samples was low, was there any indication that a particular sampling timepoint produced more positive samples than others? Perhaps pathogen presence was too low at the earlier sampling dates, but was sufficient at later dates/stages?” 

Response/. In the first submitted version, we described in detail how many samples of each developmental stage tested positive for B. glumae. Results showed that in certain sites/seasons infected samples were higher at the pre-flowering stage, but in others, higher numbers corresponded to milky and dough stages. Results did not show a specific trend, which was the reason why one of the reviewer suggested that the information did not provide enough evidence for suggesting infection dynamics related to panicle development. In the later version we removed this information.

---

## [Editor Report · Decision Letter 2]

10 May 2021

Understanding the Complexity of Disease-Climate Interactions for Rice Bacterial Panicle Blight under Tropical Conditions

PONE-D-21-01039R2

Dear Dr. Mosquera,

We’re pleased to inform you that your manuscript has been judged scientifically suitable for publication and will be formally accepted for publication once it meets all outstanding technical requirements.

Kind regards,

Prasanta K. Subudhi, Ph.D.

Academic Editor

PLOS ONE

Additional Editor Comments (optional):

Authors addressed the comments of the reviewers.

Decision-Accept
---

## [Editor Report · Acceptance letter]

17 May 2021

PONE-D-21-01039R2 

Understanding the Complexity of Disease-Climate Interactions for Rice Bacterial Panicle Blight under Tropical Conditions 

Dear Dr. Mosquera:

I'm pleased to inform you that your manuscript has been deemed suitable for publication in PLOS ONE. Congratulations! Your manuscript is now with our production department. 

Kind regards, 

on behalf of

Dr. Prasanta K. Subudhi 

Academic Editor

PLOS ONE